# EEG Thinking1 Datasets: Think-Count-Recall (TCR) and Read-Write-Type (RWT)

**Xiaodong Qu, Peiyan Liu**
Department of Computer Science
Brandeis University
Waltham, MA 02453
`xiqu,peiyanliu@brandeis.edu`

## Abstract

EEG-based Brain-Computer Interfaces (BCI) have been widely used in clinical and non-clinical research. In this paper, we present a framework to collect a large amount of EEG data with easy-to-use experiment setup, using non-invasive, wireless, and affordable hardware. Interpretable feedback generated by benchmark machine learning algorithms have been provided to the researchers and end-users. Two existing datasets are used as case studies for the framework: Read-Write-Type (RWT) and Think-Count-Recall (TCR). The goal is to inspire new machine learning approaches for decoding behavior from large-scale EEG data. The framework of experimental design, data collection, data analysis, feedback generation, and community building could pave the way towards a future when everyone can easily use BCI systems every day, similar to smartphones nowadays.

## 1 Introduction

Neural interfaces are becoming of increasing interest to industry and having large available datasets could be useful for students and researchers to tease out signals from noisy data. Brain Computer Interfaces (BCI) have been widely used for both clinical and non-clinical applications (Lotte et al. [2018a], Craik et al. [2019]), such as diagnosis of abnormal states, evaluating the effect of the treatments, helping patients with motor disabilities to move a mouse or to control a motorized wheelchair, mental workload, seizure detection, motor imagery tasks (Devlaminck et al. [2010]), BCI based games (Coyle et al. [2013]) and passive BCI. Previous research has reviewed existing datasets in the BCI field, such as Schalk et al. [2004], Lotte et al. [2007], Zhang et al. [2020], Roy et al. [2019], Miller [2019], Kaya et al. [2018], most of the datasets mentioned are collected in research labs or clinical settings with expensive medical equipment and time-consuming setup procedure, under the supervision of clinical professionals. The data collection framework we proposed allows non-expert participants to run the experiment by themselves at home, whenever they have a small amount of time, such as twenty minutes. The visual feedback generated by benchmark machine learning algorithms could help them to perform better in the future sessions.

Considering classic datasets in other domains, such as ImageNet for image classification, or MNIST for handwritten digit recognition, more data can be generated directly from the non-expert end users, and more general patterns could be recognized based on such large scale data. With the motivation to gather EEG data with a cheaper, easier and faster approach, we designed a pilot study towards building a large-scale EEG data set, for multi-class classification of user-centered tasks, generated by non-expert end-users. Results of classification with the proposed new data and machine models show a reasonable accuracy (70% to the random 20%), indicating the potential of this framework.

Submitted to the 35th Conference on Neural Information Processing Systems (NeurIPS 2021) Track on Datasets and Benchmarks. Do not distribute.

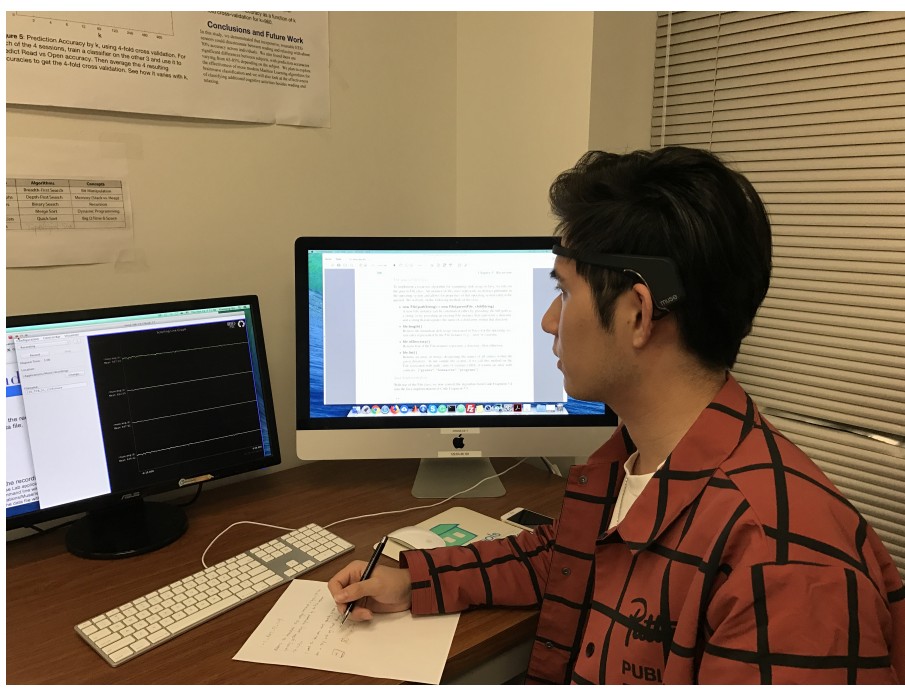

Figure 1: A researcher demonstrating the task "Write", wearing EEG headset.

This is an ongoing project and this 'Thinking1' repository currently has four datasets: Read-Write-Type (RWT, Qu et al. [2020b]), Think-Count-Recall (TCR, Qu et al. [2020a]), Python-Math (Qu et al. [2018b]), and GRE-Relax (Qu et al. [2018a]). In this paper, we use the two recent experiments, RWT and TCR, as examples to explain the approach. Details about the data collection, including how the subjects were recruited under IRB requirements, how long each session was, what kind of visualized feedback is provided to the subjects, how many EEG sessions were recorded, data cleaning, feature extraction, and research from benchmark algorithms, are in our previous papers, we also attached an updated version in the appendix section of this paper to allow readers to replicate these experiments.

## 1.1 Read-Write-Type (RWT)

Previous studies (Bird et al. [2018], Qu et al. [2018a]) demonstrated that EEG signals could successfully distinguish several kinds of cognitive tasks. Such as programming in Python vs. solving Math problems; solving Math problems (GRE) vs. solving Reading problems (GRE). These experiments focused on distinguishing different cognitive tasks, but not on whether different communication modes may also have a distinguishable impact on EEG patterns. The experiment RWT (Qu et al. [2020b]) in this data set was designed to test the hypothesis of whether AI based EEG markers could distinguish both between two modes of communication: typing vs. writing, and between three cognitive states: reading vs. copying vs. answering. The five tasks are described in Figure 2.

## 1.2 Think-Count-Recall (TCR)

Other studies (Lotte et al. [2018a], Lotte [2015], Bird et al. [2018], Qu et al. [2018a], Craik et al. [2019]) demonstrated that EEG classification was successfully used to distinguish multiple cognitive tasks. In the TCR experiment (Qu et al. [2020a]), we designed these five user-centered tasks as shown in Figure 3, abbreviated them as Think (T), Count (C), Recall (R), Breathe (B) and Draw (D). The task selection is motivated by human memory experiments such as Kahana et al. [2018].

## 2 Methods

Such datasets are suitable for machine learning due to its high dimensional and noisy nature, similar to image recognition problems. There is great potential to provide higher accuracy and more

interpretable feedback to both researchers and end-users. For example, in each data point of 1/10 second, the raw EEG data is a 4 x 5 matrix, which represents four electrodes and five frequency bands. Such twenty-dimensional data performs well enough (compare to 64 or 128 electrodes medical devices) when applied to mainstream EEG-related machine learning or deep learning algorithms.

Each session of these experiments are reproducible with twenty minutes of effort for non-experienced end-users. These human-in-the-loop experimental designs motivated by (LaRocco et al. [2020], Lotte et al. [2018b]), have several advantages. First, the tasks are selected more from the end-users, less from the researchers, similar to the smartphone usage situation now. Secondly, the role of the EEG coach can make the end-user experience much better. Last but not the least, easy-to-understand user feedback could be helpful for the end-user to reduce the noise and focus more on the designed tasks. More details in the previous papers and the appendix section of this paper.

## 2.1 Experimental Design

The experimental design is easy to adapt, and the three hundred dollars or less wireless hardware, as mentioned in Ienca et al. [2018], makes it affordable to a broader audience. For example, our research lab has expanded the experiments from just targeting less than twenty students, to a community of more than one hundred students, each of them starts with little or no computer science or neuroscience background, and usually, after at most two to three twenty-minute sessions, they can learn to how to control the noise level, and achieve the desired experimental goals with high accuracy.

The sensor hardware research and development have grown rapidly recently (Kübler et al. [2014], Tabar and Halici [2016] ), so does the trend of making it more affordable to the non-expert users. After comparing several options, such as devices mentioned in Ienca et al. [2018], we chose the Muse Headset for our experiments, with an affordable price of less than three hundred for each wireless headset. For the design of the tasks, previous research has shown deep learning works well in emotion recognition, motor imagery, mental workload, and seizure detection areas (Craik et al. [2019]), we tried learning, motor-imagery tasks, sleep, and entertainment tasks. In this study, we focused on the learning related tasks college students perform often in their daily lives.

## 2.2 Data collection

Data was collected in non-clinical settings, partly in the reserved classrooms or conference rooms in the universities, partly at the participants' home. The size of the data usually is 15 to 20 subjects, five to six sessions for each subject, each sessions varies from five minutes to twenty minutes. For example, the TCR (16 subjects) and RWT (14 subjects) experiment each includes six sessions, each session is five minutes long. Comparing with existing experiments on cognitive tasks mentioned in Craik et al. [2019], Gabard-Durnam et al. [2018], Roy et al. [2019], Pernet et al. [2019], our experimental design and data collection is easier, cheaper and faster. With twenty-minute training, most participants can generate hours of EEG recording data at home with interpretable feedback.

The non-invasive, wireless EEG headset usually needs a training session to reduce the noise level. The role of EEG coach was created to smooth the learning curve for first time end-users. The end-users and EEG coaches are fairly compensated under the IRB requirement. More details such as

**Read (R)** Subjects were asked to read a PDF file displayed on the monitor silently, the PDF file is a computer science textbook on Data Structures (Sierra and Bates [2003]).

**Write Copy (WC)** Subjects wrote on a blank white paper with a pen, copying the text from the same textbook PDF file display on the monitor. As shown in Figure 1.

**Write Answer (WA)** Subjects wrote an essay using a pen on a blank paper, answering the question: 'Why did you choose your major?'

**Type Copy (TC)** Subjects copied text from the same textbook PDF file, into a text entry box on the screen, by typing on a keyboard.

**Type Answer (TA)** Subjects typed their answers to the question 'What is your academic plan for this semester?' into a text entry box on the screen.

Figure 2: Tasks in experiment Read-Write-Type (RWT).

**Think (T)** Subjects were asked to think of several (six, seven, eight) random objects, these objects need to be independent of each other. For example, (Sun, Fish, Flower, Table, Student, Car), is a valid set, but (computer, keyboard, monitor, speaker, phone, TV) is not a valid set.

**Count (C)** Subjects counted numbers aloud, from 200 towards 0, each time subtracting by 7, e.g. 200, 193, 186, 179, with eyes open, eyes and jaws movement minimized.

**Recall (R)** Subject recalled the objects they had typed in the Think (T) task, in the correct order, if possible, and entered them in a similar text entry box with a keyboard.

**Breathe (B)** Subjects were instructed to breathe deeply with their eyes open. They were asked NOT to think about any other tasks in this experiment, or anything else except their breath.

**Draw (D)** Subjects were asked to draw the objects they thought about in the earlier task Think (T), with a pen, on a blank A4 paper. The objects text they just entered in T was displayed on the monitor, so they did not need to recall, just focus on drawing.

Figure 3: Tasks in experiment Think-Count-Recall (TCR).

IRB approval and instructions given to the participants are included in the appendix section of this paper. Each headset was connected to a mainstream personal computer through Bluetooth. We use the software package that comes with the EEG headset (Muse-io and MuseLab) to record the raw EEG data to the computers. Then the data was processed and Analyze using machine learning and deep learning algorithms. The visualized feedback is provided to the end-users, EEG coaches, and researchers to improve the next round of data collection.

Before the experiment, the EEG coach helps the end-users to understand the IRB requirements and make sure they sign the informed consent forms, then explain in detail to the end-users what they are expected to see and to do during each step. During each session of the experiments, the EEG coach leads the end-users to the experiment website to fill out the pre-experiment survey, then helps the end-users to connect the EEG headsets and conduct a test recording for one minute before the official EEG recording starts, A time-boxed online survey style guide was then used to give the end-user step-by-step prompt during the experiment, the EEG coach is there for any possible questions. After the experiment, The EEG coach makes sure the end-user fill out the post-experiment survey and help them better understand the visual feedback. Also, the EEG coach keeps track of the notes for the entire process and communicates with the researchers regularly to deal with pop-up issues and maintain a frequent-asked-question (FAQ) list.

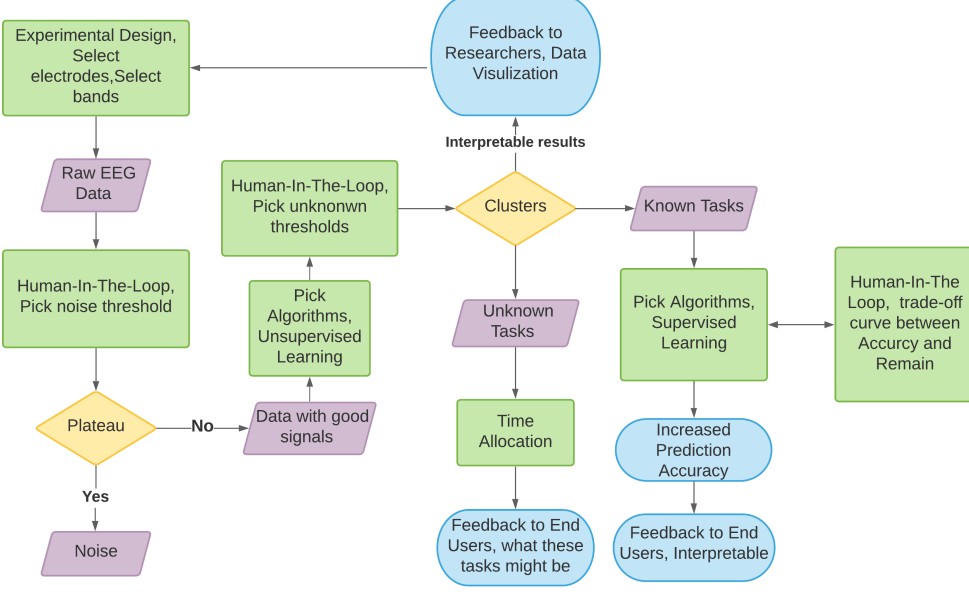

Figure 4: Data analysis framework.

## 2.3 Data analysis framework

New ways to analyze EEG data have been developed recently, such as Chevalier et al. [2020], Roc et al. [2020], Sabbagh et al. [2019], Tu et al. [2019]. In our experiments, as Figure 4 shows, the raw EEG data were first visualized to allow researchers to define the threshold to remove the noise, here we used the plateau threshold to determine whether a certain time window of signals is considered noise. Then we first used unsupervised learning algorithms, such as K-means, to cluster the data points, then we visualized the clustering result and made it interpretable to the researchers and end-user. For the designed tasks, we then used supervised learning, such as Random Forest or Long Short-Term Memory (LSTM) to predict the tasks. For the unknown tasks, we put a marker on the visual feedback to the end-user to ask what may happen during that time period. More details about these steps are in the appendix section of this paper.

By Human-In-The-Loop (HITL) it means the researcher, EEG coaches, and end-users and making data-driven decisions based on visualized feedback. The researchers bring together the existing knowledge about this experiment and help the EEG coaches and end-users to perform better in the next session. The EEG coaches guided the end-users to master how to use EEG hardware and software necessary to perform the designed tasks. The end-users learn from the researchers and the EEG coaches to use the EEG-based on BCI in their daily life as the experiments designed, and provide valuable feedback to the researchers to develop new and improved experiments.

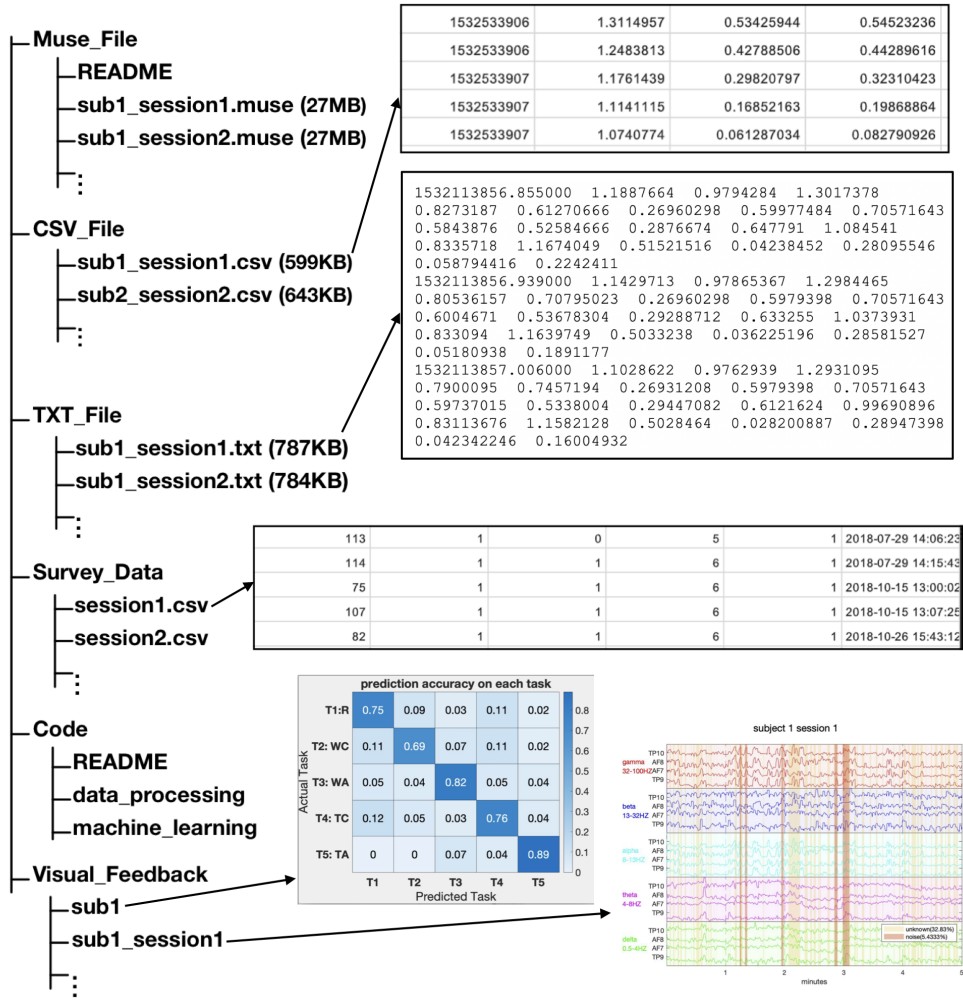

Figure 5: File structure of Our dataset

## 2.4 Data format

Pernet et al. [2019], and Nichols et al. [2017] have presented several recommended practices about EEG data formats and sharing. Our data set, as Figure 5 shows, consists of the original MUSE files, and CSV files, TXT files after pre-processing. Also, the metadata collected through Qualtrics online survey system has been included, as well as the code has been implemented for this dataset. For example, for the Read-Write-Type (RWT) experiments, for each subject, each five-minute session, there is a MUSE data file size of around 27M, and after pre-processing, the MUSE file is converted to a CSV or TXT file for further analysis, with a much smaller size of about 700K. Then there are folders of suvery metadata and related code.

## 2.5 Machine Learning applications

In this paper, we introduce a machine learning benchmark for predicting the task humans are engaged in from the EEG. We presented what machine learning and deep learning algorithms have been applied to these datasets, and suggest several recommended practices for these datasets.

For the pre-processing part, data visualization is helpful for noise detection. For the multi-class classification, ensemble methods, such as random forest, and Recurrent Neural Network (RNN), such as Long Short-Term Memory (LSTM) consistently outperformed other classifiers (Qu et al. [2020b]), we suggest using them as benchmark algorithms. Building on top of that, we proposed our algorithm, Time-Continuity-Voting (TCV, Qu et al. [2020a]), which achieved the highest prediction accuracy for these datasets. More details are in the appendix section of this paper.

## 2.6 Community Forum and further support

We established an online forum for the community who works on these datasets, including researchers, EEG coaches, end-users, and clinical professionals. Due to our IRB requirements, this forum is invitation-only at this time. Through our BCI forum, we connected to three computer science labs, two neuroscience labs, two clinical research labs, and two hospitals during the last three years, as well as get more than a hundred undergraduate students involved as experiment participants, eight of them later became EEG coaches.

We held discussions on how to improve EEG experimental design and dataset development. Further support on how to explore the potential of such an EEG-based BCI system is encouraged based on community members' availability. Also, we are presenting these research papers and this forum to more college students in the computer science and Neuroscience courses we lectured each semester.

Participating in the existing BCI community and bridging our own small EEG-based BCI community to a broad network is also an important direction.

## 2.7 Availability and Ethical considerations

To make sure these datasets would be used ethically and responsibly, we adapted several recommended practices of sharing BCI data, such as Gabard-Durnam et al. [2018], Pernet et al. [2019]. According to our IRB requirement, these data sets are available upon written request, we review the request to make sure it is coming from a reputable research institution and the requester is willing to sign a Non-Disclosure Agreement. Previous studies have reviewed freely available EEG datasets, such as Zhang et al. [2020], Roy et al. [2019], Miller [2019], Kaya et al. [2018], Craik et al. [2019], we are amending our IRB to find acceptable ways of data anonymization to share it more freely.

# 3 Results

We develop feedback for different user roles. For example, the figures that compare different end-users or different machine learning algorithms are more for the researchers, optional for the end-users. Here are some sample feedback figures we provide to our researchers, EEG coaches and end-users.

 **3.1 For Researchers**

 For cross-subject comparison, as Figure 6 shows, although there are individual differences, the task
 prediction accuracy is reasonably high. Together with Figure 7, (both figures X-axis is subject id
 ordered by prediction accuracy), we observed the noise and unknown tasks vary across different

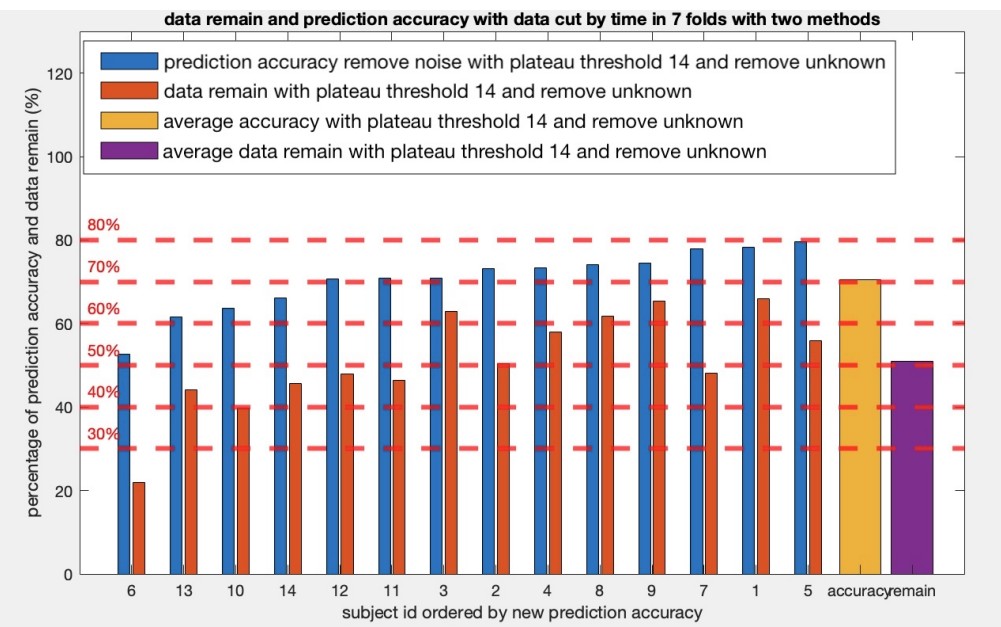

Figure 6: Experiment RWT: task prediction accuracy and data remain.

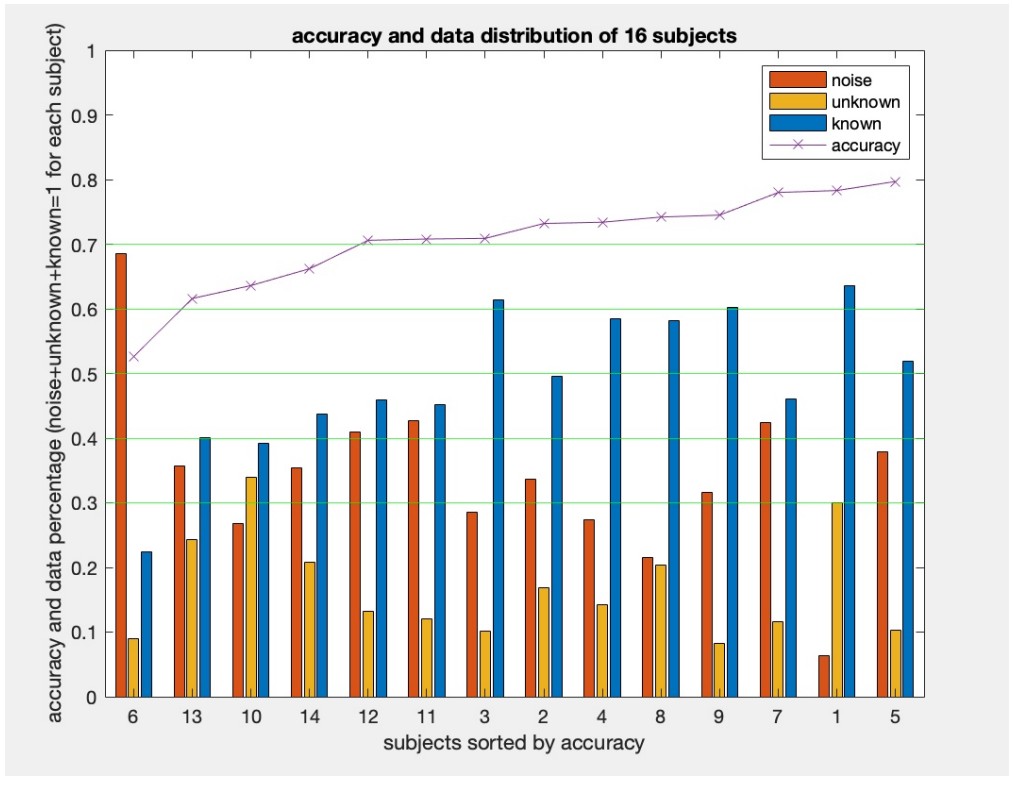

Figure 7: Experiment RWT: noise, unknown, and known tasks percentage.

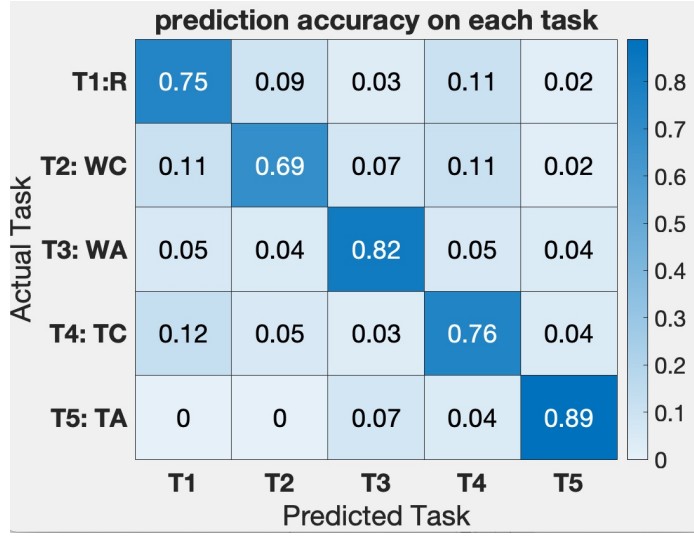

Figure 8: Diagonal Accuracy

subjects. Thus end-user training is necessary for better controlling the noise and unknown tasks. The role of EEG coach is created for this purpose.

## 3.2 For End-Users

Figure 8 shows for subject one in experiment RWT, how the accuracy of each task is predicted over all six sessions. This feedback may guide the further task selections. Each individual has a unique task set that is easy to be recognized with this EEG-based BCI experimental design and data collection framework. Thus it has the potential to be used as personal EEG fingerprint. Figure 9 shows the noise

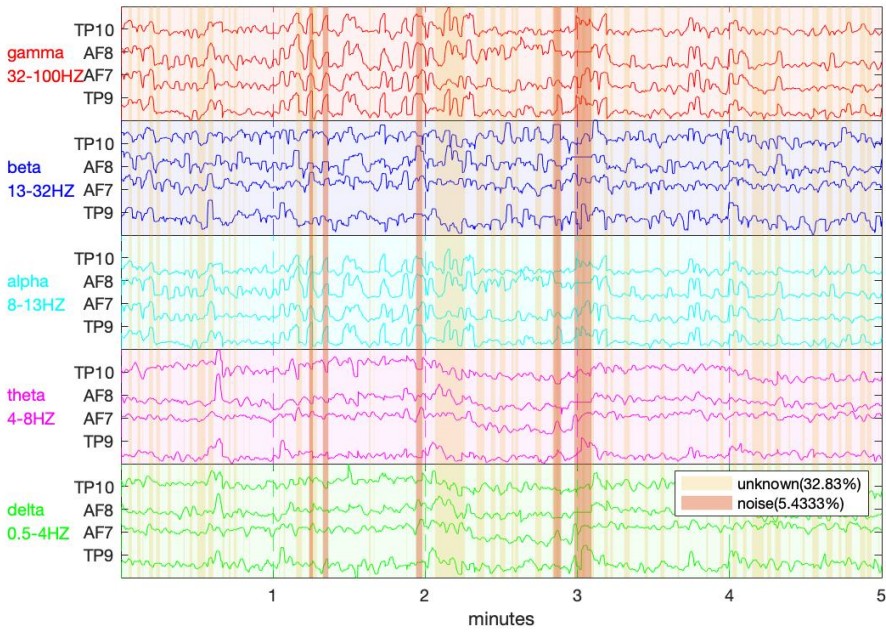

Figure 9: Noise, unknown, and known tasks, experiment RWT: subject one, session one.

188 and unknown task locations in each session, such feedback is helpful for the end-user to reflect what
189 happened around a certain time spot.

190 ## 4 Discussion

191 The main goal for this paper is to provide a framework of experimental design and data collection to
192 gather EEG data through cheap means and non-expert participants. Intepretable feedback generate by
193 benchmark machine learning algorithms can speed up this process. Comparing with the traditional
194 data collection methods, as mentioned in Craik et al. [2019], Gabard-Durnam et al. [2018], Roy
195 et al. [2019], Pernet et al. [2019], our approach is faster and cheaper to gather more EEG data with
196 non-expert participants. Our efforts are made toward the future of everyone can use EEG-based BCI
197 in their daily life, similar to the current everyday usage of smartphones. Although the limitation
198 of sensory accuracy will remain for a while, the research related to the non-invasive BCI shows a
199 growing potential to reach out to non-expert end-users.

200 The datasets we present are an early exploration of how to map the healthy subjects' daily activities to
201 their personal EEG signal patterns. Based on the currently available sensory hardware, tasks without
202 too much moving or talking could be a good start. A unique role of EEG coach could be helpful in
203 such experiments to encourage more end-users to get involved in such experiments. The short-term
204 goal of these datasets is to inspire new machine learning approaches for decoding behavior from
205 EEG.

206 ### 4.1 Future Work

207 Neural interfaces are becoming of increasing interest to industry and having large available datasets
208 could be useful for students and researchers to tease out signals from noisy data. As non-invasive
209 neural recordings become ubiquitous, there is a greater need for such algorithms and datasets. We
210 will continue to focus on developing a framework to make it easier for non-expert end-users to use
211 EEG-based BCI. Our short-term exploration includes developing more specific role sets for the BCI
212 research and development framework, with the emphasis on the role of EEG coach and an online
213 EEG experience community. The impact of continuous feedback to end-users is also a topic we
214 are working on. Also, the idea of step out of the lab to home, starting with encouraging end-users
215 to record EEG during tasks of her/his choice as many times as possible at home, is an interesting
216 direction we are heading to.

217 ### 4.2 Broader Impact

218 This approach could contribute to the building of a large-scale EEG dataset using low-cost tools and
219 simple experimental settings at home. Our framework could be illuminated to a broader audience
220 of other time-serious human sensory data collection. After all, brain signals are just one type of
221 sensory health signals, the development of wearable devices are expanding rapidly to provide more
222 perspective about human health information from both real-time monitoring and afterward data
223 analysis.

224 Together with other human sensory data, EEG-based BCI has the potential to significantly change
225 the ways of human interaction with the rest of the world, including both other individuals, and all
226 the technology devices we developed. The human brain is a type of high-speed neural network, and
227 the current AI-enhanced internet is also a high-speed network, how to connect the two high-speed
228 networks could be an interesting long-term research direction. Our pilot study of quickly gathering
229 large-scale EEG data could be a baby step moving towards this direction.

230 ## 5 Conclusion

231 In this paper, we present a framework to gather large-scale EEG data through cheap means and
232 non-expert participants, including experimental design, data collection, data analysis, and community
233 building approaches. Two existing datasets are used as case studies for the framework: Think-Count-
234 Recall (TCR) and Read-Write-Type (RWT). This could be a building block towards the future of
235 everyone using non-invasive, wireless, and affordable BCI systems every day, similar to current
236 smartphone usage for the general non-expert population.

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
