# OpenReview forum: "EEG Thinking1 Datasets: Think-Count-Recall (TCR) and Read-Write-Type (RWT)"
_NeurIPS.cc/2021/Track/Datasets_and_Benchmarks/Round1 — Submitted to NeurIPS 2021 Datasets and Benchmarks Track (Round 1)_

### Official Review · Reviewer_TfwL · 2021-06-30
**EEG Dataset with insufficient clarity and documentation**

**Rating:** 3
**Confidence:** 4

**Strengths:**

# Significance of the contribution.
Neural interfaces are becoming of increasing interest to industry and having large available datasets could be useful for students and researchers to tease out signal from noisy data. This research is well situated with respect to that goal.




**Weaknesses:**


# Significance of the contribution,
The work is unfinished and poorly documented, making it difficult to evaluate and follow. It is not clear what their dataset consists of (tasks and subjects), what algorithm was used for benchmarks, or how to get access to the dataset. The paper is poorly written, and it is not clear how many subjects and activities are included, and their extent. The paper does not discuss other EEG/ECoG datasets that may be relevant in the field.


# Relevance to the broader research community.
Such a dataset, if of sufficient size, scope, and quality, could be of interest to the Neurips community. However I do not believe this dataset goes beyond already available datasets for inspiring algorithm design from brain signals. I am not sure what the novelty is.

 # Accessibility and accountability,
The authors do not include information about research with human subjects in the author statement, e.g. discussion of wages, participant links, IRB approvals, or instructions given to patients. Numerous other fields in the checklist (eg. amount of compute, code reproducibility) were marked as 'yes', but I see no documentation of processing power.



**Additional Feedback:**

Even were the clarity issue with this dataset resolved, I am not convinced it is of sufficient breadth and interest for this conference.

**Clarity:**

No. I found the introduction and motivation sections easy to follow, but much of the rest was not. The figures were undocumented, discussed out of of order, and figure captions were left unfinished with author comments (Figure 4). The results section was difficult to follow and the methods behind the benchmarks are not clear. The reference formatting occludes the text, and the instructions for the checklist and submission were not commented out.

**Correctness:**

The dataset file structure is documented, but it is not otherwise clear what the data consists of, how many subjects there are, how many tasks and sessions there are, or if there is a metadata. It is not clear what the benchmark is, and if the design and methods of the algorithm are appropriate.

**Documentation:**

There is not sufficient detail on data collection, availability and maintenance. There is a description of the file structure (Figure 5), but it is unclear what each of the files is (raw data? processed data? what is the information in the qualitrics survey?). There is not available URL to the dataset, or a hosting, licensing, and maintenance plan, or a way to support reproducibility.

**Ethics:**

There are eCog and likely EEG datasets (e.g. Miller, K.J. et al. Nature Human Behavior 2019) that are freely available. The authors' may consider amending their IRB to find an acceptable means of anonymizing the data to share it more freely.

**Relation To Prior Work:**

No. There is a discussion of past work on the specific tasks used, but there is not a discussion of different datasets for EEG and ECog. This is especially problematic, because such datasets are available in the BCI field (e.g. Miller, K.J. A library of human electrocorticographic data and analyses. Nat Hum Behav 3, 1225–1235 (2019); Schalk, G., McFarland, D.J., Hinterberger, T., Birbaumer, N., Wolpaw, J.R. BCI2000: A General-Purpose Brain-Computer Interface (BCI) System. IEEE Transactions on Biomedical Engineering 51(6):1034-1043, 2004; the Berlin BCI Competition; Kaya, M., Binli, M., Ozbay, E. et al. A large electroencephalographic motor imagery dataset for electroencephalographic brain computer interfaces. Sci Data 5, 180211 (2018)). There are likely available datasets in the more cognitive-EEG field, but there are not discussed, making it difficult to evaluate the novelty  of the approach.

**Summary And Contributions:**

Update: 7/20/21 . Thank you for the revised submission. I do think that some of the missing information has been added to the manuscript, and I have improved the score to a 3, but I still think that the manuscript has a ways to go before being of suitable quality and documentation level for acceptance. There is a basic confusion over the exact nature of the dataset, and whether what is being presented is a dataset or a framework. I'm not sure a framework would be suited to this track.

This submission introduces a dataset of EEG signals in humans performing different activities, intended for multi-class classification of tasks such as reading, writing, and typing. The authors introduce a machine learning benchmark on predicting the task humans are engaged in from the EEG. The ultimate goal of these datasets is to inspire new machine learning approaches for decoding behavior from EEG. As non-invasive neural recordings become ubiquitous, there is a greater need for such algorithms and datasets. I believe the dataset is from 20 EEG contacts sampled at 10 Hz, from 14 subjects, in 2-10 different tasks across six sessions, but am not absolutely sure due to the manuscripts clarity. Subjects are trained to use the EEG beforehand. The data is stored, in standardized EEG formats, but the nature of the metadata is unclear.  The data is only offered via written request.

---

> ### Author Response · Authors · 2021-07-15
> **Improved novelty, relation to prior work, documentation and availability. Thanks.**
>
> Thank you so much for the detailed and constructive feedback.
>
> One of the big changes we have made is to add the details about data collection and data analysis.  First, we cite our previous papers about each of these individual experiments, TCR and RWT,  then we also include an updated version of these details in the appendix section, to make it easier for the readers to reproduce the experimental results.  Also, the information about research with human subjects has been added to both the 'Introduction' section and the appendix section.
>
> Another issue is about the novelty, we are aiming to provide a framework that non-expert participants can gather EEG data in twenty minutes at home, with devices cost less than three hundred dollars. And the interpretable feedback generated by the benchmark machine learning algorithms could speed up this process. The role of the EEG coach and the building of an online community is also helpful to establish such large-scale datasets. We explained this more in the 'Introduction' and 'Discussion' sections. We further explained our idea of non-expert participants provide a large amount of EEG recording at home when they have twenty-minute spare time, with initial help from EEG coaches, and automatically feedback from benchmark algorithms.
>
> For the relation to prior work, we appreciated the suggestion of adding the three additional studies, we included all of them in the 'Introduction' Section, to further explain what's the differences between our framework and the existing datasets. Also, Miller_2019 and other related papers are really helpful to amend our IRB and find better ways to make our datasets available more freely, we added them to the "Availability" and 'Future Work" subsection.
>
> last but not least, we reorganized the storytelling and trying to solve the clarity problem by only address one issue in this paper and move the other ideas to future work or appendix.
>
> Thanks again for your supportive feedback to guide us to improve this paper.

---

### Official Review · Reviewer_Unbv · 2021-07-03
**Proposes framework for building large-scale EEG datasets, but significance of contributions is unclear**

**Rating:** 3
**Confidence:** 4
**Correctness:** Yes

**Strengths:**

EEG-based BCI systems have uses for many applications. Standardizing datasets and enabling the creation of large-scale datasets is likely critical for their development.

**Weaknesses:**

Generally unclear what the significance and contribution of this paper is.

**Additional Feedback:**

Formatting of citations makes it difficult to read.
Caption of Fig. 4 seems to be incomplete
Hard to tell how to read and what to take away from Fig. 6


**Clarity:**

The organization and clarity of writing could be improved. In general, the writing and formatting seems a bit unpolished. Also see "additional feedback" for some specifics.

**Documentation:**

No, availability and maintenance are not discussed.

**Ethics:**

No.

**Relation To Prior Work:**

Lacks significant discussion about the proposed framework’s relationship to prior work, it’s exact contributions, etc. For example, Sections 1.1 and 1.2 could use more context and explanation. It is not clear what is prior work and what (if any) is being proposed.

**Summary And Contributions:**

This paper presents a framework for creating EEG datasets that can support BCI systems. Core components of the pipeline include experimental design, data collection, data analysis, and community forum. Two existing datasets are used as case studies for the framework.

---

> ### Author Response · Authors · 2021-07-15
> **Improved significance, clarity of writing, relation to prior work and documentation. Thanks.**
>
> Reviewer Unbv:
>
> Thanks for your feedback.
>
> About the significance and the contribution of this paper. We added in the 'Introduction' and 'Discussion' sections. Mainly it is gathering large-scale EEG data with an easy experiment set up and low-cost (<$300) headsets. Make it easier for non-expert participants to use it at home, and provide interpretable feedback generated by benchmark machine learning algorithms, could speed up the data collection process.  The role of EEG coach and community building are also helpful in this framework.
>
> About the organization and clarity of writing, we reorganized the structure and double-checked the writing, with a special focus on the storytelling, figures, and citations. Thanks for the details in the additional feedback section.
>
> About the relations to prior work,  we added more discussion in the 'Introduction',  'Method' and 'Discussion'  sections to explain and compare.
>
> For the documentation, we cited our previous papers about the data collection details, also we add more details in the appendix section to make it easier for the readers to reproduce the experiments.

---

### Official Review · Reviewer_NEna · 2021-07-05
**The paper explores an important topic of gathering EEG data through cheap means and non-expert participants. However, I find the details on the dataset statistics , dataset collection and benchmarks models are not sufficient to allow readers to replicate these experiments. Also, the writing is not clear and grammatically correct. Hence, I think this manuscript needs a major revision before it is ready for publication.**

**Rating:** 3
**Confidence:** 4

**Strengths:**

* The authors pursue an important research direction where they try to collect a large amount of data using cheaper tools and simple experimental setting. This type of approach if successful  could help to quickly gather large scale data in medical community
* Results show that these data achieve a reasonable accuracy ~ 70%
* The data will be provided upon request by validating the institution and research area of the requesting person

**Weaknesses:**

* While the motivation to gather medical data with cheaper, easier and faster approach is important the details provided in the paper are not enough to replicate the same data collection process. For example,
    * During data collection, how are the volunteers chosen, how long was each session, what kid of visualized feedback is send to the participants, how many EEG cycles were recorded, etc.
   * Data analysis: The details of  human-in-the-loop process is not provided. How should the EEG coach guide the end-user?
  * What are the details of the ML model? Was the data preprocessed before using it in a models. The dataset size is not clear. What is the threshold and what value of K was the best?

* The clinical significance of the present experiments and task are not explained in the paper so it is difficult to access how important the designed experiments are

* There is no comparison to data collected in clinical/research setting

* The writing is not clear and there are many grammar issues.
   * there is no space around () at multiple places
   * Cross line figure [cite Figure]

**Additional Feedback:**

I think the paper could be improved by adding quantitative details on dataset statistics and dataset collection process. The authors could also add additional ML/DL models to conduct these experiments and improve the baseline numbers.

Also, the paper needs to be revised for clarity, consistency and grammar.

**Clarity:**

The paper is not clear. It has many grammatical and minor issues. It needs to be revised throughout for clarity. Below I point out some issues:
* Think(T), count(C), recall(R) -> Think (T)
* Incorrect sentence: These datasetsQu et al. [2020b] suggest a future direction
* comma and spaces: Qu et al. [2020a]LaRocco et al. [2020]Lotte et al. [2018b].
* Can you please change qualitative description with quantitative numbers:low-cost -> what was the cost compared to clinical equipment's, interpretable feedback -> what feedback was provided?
* Please provide the link to the online forum
* Use sentence case in all figure captions


**Correctness:**

* The main contribution of the paper is to provide an experimental design   and data collection pipeline to users to easily collect EEG data. However, the experiment steps are not clear which will make it difficult for reader to design and conduct other similar experiments?
* How are the participants picked? How important is it for the participants to be college students?
* Will continuous feedback to participants during the experiments impact the outcome of the experiments in anyway? Also, do you discard initial data points when the participants were not experienced?


**Documentation:**

As described in above comments, I think the details are lacking both on the dataset statistics and on data collection process.

**Ethics:**

* Was appropriate approval and permission sought for publishing Figure 1 of a participant. * * Were the participants and EEG coach fairly compensated?

**Relation To Prior Work:**

* The authors provide some references and connect their work to these references at the beginning of each section.
* It is not clear how the proposed TCR and RWT compare to existing experiments on cognitive tasks. What devices are used by present works could be added.
* The size and format of this data in comparison to existing data is not available but it could be helpful to users.
* What deep/machine learning models do existing works use

**Summary And Contributions:**

* This paper presents  a pilot study to build large scale EEG data that can be used for multi-class classification of use-centered tasks. The paper presents experiment design and data collection process for two EEG based BCI experiments:
     * Read-Write-Type:  typing vs. writing
    * Think-Count-Recall: cognitive states like copying vs. answering vs. reading

* The authors propose the use of low-cost wireless hardware: the Muse headset
* The authors developed an online forum for discussion and use of this dataset
* Results of classification with the proposed new data and machine/deep learning models show a reasonable accuracy indicating the potential of this data collection method

---

> ### Author Response · Authors · 2021-07-15
> **Added details of the framework, documentation, quantitative numbers and improved writing clarity.  Thanks.**
>
> Thank you so much for the detailed and constructive feedback. It really helps us to shape our ideas and improve our writing.
>
> We started by adding the details about data collection and data analysis.  First, we cite our previous papers about each of these individual experiments, TCR and RWT,  then we also include an updated version of these details in the appendix section, to make it easier for the readers to reproduce the experimental results. The previous papers are focusing on the individual experiments or a single algorithm, this paper is more about a generalizable framework and benchmark approach.
> The details of visual feedback to different roles of researchers, EEG coaches, and end-users are also first-time presented in this paper. The community building part has also contributed to the large-scale data gathering process.  Human-In-The-Loop is another topic we are interested in, but due to the page limit, we considered moving it to another paper to further explain the details, at the same time, attach enough details in the appendix section of this paper for reproducibility.  Also, what machine learning/deep learning algorithms have been implemented have been documented in both the 'Method' section and Appendix.
>
> Thanks for identifying the impact of continuous feedback on participants, that's one topic we are really interested in and are currently collecting more data during this summer, so far the results imply the learning effect is more obvious after the EEG coach explains the automatically-generated feedback to the end-users.  So it still involves human learning and coaching, not purely learning based on automatic feedback. We put it in the future work section and will continue working in this direction.
>
> For the relation to prior work, we appreciated the suggestion of adding additional studies, we included more comparisons with other clinical datasets in the 'Introduction' Section, to further explain what's the differences between our framework and the existing datasets.  For the equipment we used, we added more citations about what other options we compared and why we pick this one, also we briefly mentioned the cost of money and time for our experiments, in comparison with other cognitive task experiments.  We also added the appropriate approval and permission sought for publishing Figure 1 as a participant, and the details about how we recruited and compensated the participants and EEG Coaches in both the 'Method' section and Appendix.
>
> Another question we have is the difference between clinical and non-clinical use, if we are aiming to motivate non-expert end-users to use it at home for everyday tasks, is this still considered as clinical use?  Our long-term goal is to make the non-invasive BCI user experience more like a smartphone or smart watch experience,  rather than a clinical device, does it make more sense for us to use the 'non-clinical term to describe our approach? Thank you so much.
>
> About the writing clarity, we really appreciate your kind support about the details. we changed the spaces around () throughout the paper. We revisited all the figures, and the text related to the figures. We proofread to double-check the grammatical and minor issues.  We changed the format of citations, the use of comma and spaces.  Also, we add quantitative numbers, low-cost -> less than three hundred dollars,  interpretable feedback -> examples visual feedback figures, and the explanation. For the link to the online forum, because according to our previous IRB, it is still invitation only. We added in our future work that amending the IRB to make the datasets available more freely.
>
> Thanks again for your detailed and supportive feedback to guide us to improve this paper,  and more broadly to our research. I personally really enjoy working on each section of your replies and I can feel the personal growth by getting each issue resolved.  I really appreciate that.

---

### Decision · Program_Chairs · 2021-07-26

**Decision:**

Reject

**Comment:**

This paper is about a dataset for EEG-based BCI tasks.  The reviewers found the topic interesting, but found that the dataset/paper was not sufficiently well-motivated or documented.